# Contemporaneous SARS-CoV-2-Neutralizing Antibodies Mediated by N-glycan Shields

**DOI:** 10.3390/v15102079

**Published:** 2023-10-12

**Authors:** Leili Baghaie, Fleur Leroy, Mehdi Sheikhi, Abdollah Jafarzadeh, Myron R. Szewczuk, Abdolkarim Sheikhi

**Affiliations:** 1Department of Biomedical & Molecular Sciences, Queen’s University, Kingston, ON K7L 3N6, Canada; 16lbn1@queensu.ca; 2Faculté de Médecine, Maïeutique et Sciences de la Santé, Université de Strasbourg, F-67000 Strasbourg, France; fleur.leroy@etu.unistra.fr; 3Department of Immunology, School of Medicine, Dezful University of Medical Sciences, Dezful 64616-43993, Iran; mehsha_2012@hotmail.com; 4Faculty of Medicine, Kazeroon Azad University, Kazeroon 14778-93855, Iran; 5Department of Immunology, Medical School, Kerman University of Medical Sciences, Kerman 76169-13555, Iran; jafarzadeh14@yahoo.com

**Keywords:** COVID-19, SARS-CoV-2, original antigenic sin, cross-reactive lymphocytes, N-glycan

## Abstract

Mutations and the glycosylation of epitopes can convert immunogenic epitopes into non-immunogenic ones via natural selection or evolutionary pressure, thereby decreasing their sensitivity to neutralizing antibodies. Based on Thomas Francis’s theory, memory B and T cells induced during primary infections or vaccination will freeze the new mutated epitopes specific to naïve B and T cells from the repertoire. On this basis, some researchers argue that the current vaccines derived from the previous strains of the SARS-CoV-2 virus do not increase immunity and may also prevent the immune response against new epitopes. However, evidence shows that even if the binding affinity is reduced, the previous antibodies or T cell receptors (TCRs) can still bind to this new epitope of the Beta, Gamma, and Delta variant if their concentration is high enough (from a booster injection) and neutralize the virus. This paper presents some convincing immunological reasons that may challenge this theory and argue for the continuation of universal vaccination to prevent further mutations of the SARS-CoV-2 virus. Simultaneously, the information presented can be used to develop vaccines that target novel epitopes or create new recombinant drugs that do not lose their effectiveness when the virus mutates.

## 1. Significance

One crucial aspect of SARS-CoV-2 is the ability of the virus to rapidly mutate and create antigenically distinct strains. Changes in the amino acids and glycosylation or deglycosylation of sites create new epitopes by changing the previous epitopes. These established new epitopes form novel N-glycan shields that can mediate other contemporaneous SARS-CoV-2-neutralizing antibodies.

## 2. Introduction

### 2.1. Contextual Framework

The Original antigenic sin (OAS) theory, described in 1960 by Thomas Francis, states that the immune system preferentially uses immunological memory based on a previous infection when encountering a second, slightly different version of that foreign pathogen. This leaves the immune system “trapped” by its first response to each antigen and unable to mount potentially more effective responses during subsequent infections. Based on this theory, memory B and T cells induced during infections or vaccinations with the primary variant of the pathogen will freeze the new mutated epitopes’ specific naïve B and T cells (cross-reactive memory against specific naïve B or T cells) from the repertoire [1]. Some researchers argue that if a booster dose of the SARS-CoV-2 vaccine from the primary variant is administered, even when there are common epitopes between the two variants, the immune response against the new uncommon epitopes will be prevented, thus failing to enhance immunity [2]. Garrity et al. [3] introduced this process in relation to the human immunodeficiency virus (HIV) and coined it decotope, or immune decoy epitopes. This process was defined as a shift from immunodominant epitopes to a limited pool of neutralizing antibodies providing low protection [3,4]. Here, we argue against this notion and present several ways in which the immune system can still mount a response against mutated variants.

### 2.2. Perspective

Suppose a primary antigen with several epitopes enters the body. The epitopes are divided into several categories: exposed immunogenic epitopes, hidden immunogenic epitopes, exposed non-immunogenic epitopes, and hidden non-immunogenic epitopes. The exposed immunogenic epitopes (linear and conformational) are expected to naturally stimulate the cellular and humoral immune system and induce T and B cell memories. Similarly, the hidden immunogenic linear epitopes will stimulate the humoral and cellular immune systems by producing antigen-specific antibodies (Ab1) during the first immune reaction following exposure to the antigen. In the case of SARS-CoV-2, mutations might have several consequences: the exposed immunogenic epitopes may change so that the antibodies produced against these epitopes no longer can neutralize the virus, and the virus will remain pathogenic.

Furthermore, the production of anti-idiotype antibodies (Ab2) can be induced, specifically targeting Ab1 to inhibit its action by forming immune complexes. Paratopes of Ab2 can mimic antigens by binding to Ab1. Moreover, their structural similarities enable them to bind to the specific receptor of the original antigen and induce agonist or antagonist cell signaling in the cells targeted by the virus. This mechanism mimics the pathological reaction and triggers a long-term response after the first contact. Characterizing the action of Ab2 would help us to understand several adverse effects during SARS-CoV-2 infection or vaccination [5].

### 2.3. Immunologic Reasons Challenging OAS Theory

After mutation due to changes in amino acids or glycosylation (or deglycosylation), new epitopes are created by changing the previous epitopes. Even if the binding affinity is reduced, the previous antibodies or T cell receptors (TCRs) can still bind to this new epitope if their concentration is high enough (from a booster injection) to neutralize the virus. This has been demonstrated based on the neutralization of new variants of the SARS-CoV-2 virus (Beta, Gamma, and Delta) through convalescent and post-vaccination sera with high titers [6]. On the other hand, if the binding affinity is too low and the mutated epitopes constitute critical residues, or new glycosylation/deglycosylation occurs on the residues, improving access to the ACE2 receptor, the affinity of the antibodies will drop significantly [7,8,9]. The more specific naive B and T cells in the repertoire will also gain access, bind to the mutated epitopes, and be stimulated. Evidence shows that changes in the glycan components of the SARS-CoV-2 S protein can profoundly influence the epitopes targeted by neutralizing antibodies [10,11,12,13] (Figure 1A).

Another argument is that mutations and glycosylation can convert immunogenic epitopes into non-immunogenic ones via natural selection or evolutionary pressure, decreasing their sensitivity to neutralizing antibodies [14]. Unlike bacteria, in which glycans are encoded by the bacterial genome and treated as “nonself” epitopes by their corresponding hosts, viruses take advantage of the host cell machinery for glycosylation. Generally, they are decorated with the “self”-glycans. These “self”-glycans are generally considered as a strategy for escaping the host immune response [15,16]. Even though the previous antibodies or TCRs do not bind to the converted epitopes, the new epitopes do not stimulate more specific naive B cells and T cells from the repertoire. However, we cannot rely on this antigenic sin for previous B or T cell memories.

Glycans disrupt the immune response and promote immune escape by inhibiting recognition by antibodies and immune cells. In influenza, the severity of the disease depends on the glycosylation profile of surface proteins such as hemagglutinin (HA) from past exposure. Infection with a highly glycosylated variant induces a deficient adaptive response with a lack of neutralizing antibodies and a robust T response. In the event of reinfection with a low-glycosylated variant, antibody-mediated neutralization is significantly reduced, and the disease is more severe. This explains why hosts previously infected with highly glycosylated seasonal H1N1 viruses develop severe lower respiratory tract disease due to the pandemic H1N1 strain [17].

Furthermore, some evidence indicates that some new glycosylation on the SARS-CoV-2 spike glycoprotein not only shields the immunogenic epitope but also, at the same time, may change the S binding affinity to ACE2 (Figure 1B). Zhao et al. suggested the essential roles of glycosylation in mediating receptor binding, antigenic shielding, and potentially, the evolution/divergence of these glycoproteins [18]. This process could be necessary for vaccine stabilization via glycosylation and could be considered when manufacturing future SARS-CoV-2 vaccines [19].

The professional antigen-presenting cells (APC), B lymphocytes, also present an opportunity to refute the OAS theory. Any antigen it recognizes can present its different epitopes and the epitope that specifically binds to its receptor, can bind to T helper lymphocytes in the MHC class II cleft (Figure 1C). The glycan-specific B lymphocytes can be helped by the previous memory T cells specific to epitopes other than the mutated (glycosylated or deglycosylated) epitopes of the SARS-CoV-2 spike. Here, it can be concluded that using the boosters of existing vaccines can at least induce memory T cells specific to epitopes other than the mutated epitopes of the SARS-CoV-2 spike (common epitopes of the primary and the mutated variants). These non-mutated T-dependent epitopes can help B lymphocytes to mount a specific response to new SARS-CoV-2 variants. Therefore, not only do the previous B lymphocytes fail to prevent the stimulation of B lymphocytes specific to new epitopes, but they also involve T helper cells in order to induce the response of new B lymphocytes specific to the mutated epitopes [20].

After mutation, the change in the epitopes mentioned above may lead to a situation where the previous vaccine-induced antibody’s binding to their adjacent epitopes increases their binding affinity to ACE2. As Liu et al. reported, several antibodies bind to parts of the SARS-CoV-2 N-terminal domain (NTD), increasing its binding to the ACE2 receptor [21]. This is an FcR-independent antibody-dependent enhancement (ADE) of the infection induced by the previous SARS-CoV-2 variants. Other evidence indicates that individuals vaccinated with antigens against Wuhan-Hu-1 and then infected with the Alpha or Delta variants have a relatively decreased response to variant-specific epitopes compared to unvaccinated individuals who become naturally infected [22]. Unvaccinated individuals were found to have immunity against SARS-CoV-2 only if they had a prior infection. Unvaccinated individuals who were naturally infected had an 85% lower risk of contracting COVID-19 than those who were not naturally infected [23]. This was seen in both mild and severe disease cases. This could be due to the Fc-independent ADE, as there is no evidence that previous memory B lymphocytes prevent the stimulation of B lymphocytes specific to new epitopes [22,24]. Although these Fc-dependent and -independent ADE antibodies were produced through the previous virus variants’ stimulation (i.e., very few vaccine platforms or natural infection), the leading cause of this problem was the mutation of the virus. So, the mutated antigen is the culprit, not the original antigen (Figure 1D).

Mattiuzzi and Lippi [25] conducted a review of the analyses of COVID-19 vaccine efficacy in older persons who received the second booster compared to unvaccinated people and those who received only a single COVID-19 vaccine booster. The second vaccine booster maintained high effectiveness against adverse COVID-19 outcomes such as hospitalization, intensive care unit admission, and death (i.e., between 77 and 86%) and also showed an efficacy around 10% higher than the single booster. The efficacy of the second vaccine booster declined over time, decreasing by 33–46% when assessed at >120 days from administration. The results of these ad interim analyses of the ongoing Italian nationwide COVID-19 vaccination campaign suggested that regular boosting with COVID-19 vaccines may be advisable for older persons [25].

The SARS-CoV-2 S glycoprotein (SGP) is a trimeric class I fusion protein that comprises two subunits, including the S1, the receptor-binding subunit attached to the S2, the fusion-mediating subunit, such that the complete entity is a trimer of S1–S2 heterodimers [26]. The S2 subunit is anchored to the virus via a membrane-spanning domain. Once assembled and processed by a protease within the cell, the fusion protein is maintained in a metastable state known as the prefusion conformation [26]. Usually, the receptor-binding subunit (S1) overlays its fusion-mediating counterpart (S2) and temporarily locks it into an energetically unfavorable conformation. When the receptor-binding subunit engages ACE2, its structure alters so that it releases the S2 subunit to undergo profound conformational changes, revealing a hydrophobic region at the N-terminus of the S2 subunit. This region can be inserted into the cell membrane, creating a linkage protein between the cell membranes and the virus. The release of these energies is sufficient to pull the two membranes together so as to allow them to fuse. The S2 subunit is now in its post-fusion conformation. As a vaccine candidate, the mutated SARS-CoV-2 SGP can produce epitopes for virus-neutralizing antibodies (NAbs). When used as an immunogen, it induces NAbs that, in turn, will bind to the same protein on the virus surface, impairing its functions and neutralizing its infectivity. The relevant NAb epitopes are optimally displayed or observed only when the trimer is in its conformational perfusion.

Conversely, post-fusion or other aberrant protein conformations induce primarily non-neutralizing antibodies (non-NAbs) with no or limited protective capacity. In other cases, non-NAbs can even be harmful. When expressed as a recombinant protein, the SARS-CoV-2 SGP trimer can undergo spontaneous conformational changes and decay into its post-fusion form, inducing non-NAbs. These non-NAbs can be categorized as Fc-antibody-dependent infection enhancement antibodies produced by the previous vaccination, and the induction of such non-NAbs results in immune complex (IC) deposition and complement activation. These IC deposits have been linked to COVID-19 disease severity and many of the long-term consequences of SARS-CoV-2 infection such as arthritis and vasculitis [27]. This viral persistence based on the enhanced interference of non-NAbs has also been noted in HIV. The destruction of CD4+ T cells, downregulation of MHC molecules, and the establishment of latent viral genomes are often linked to immune evasion, disease progression, and non-NAbs [28]. Overcoming this problem requires the application of protein-engineering stabilizing methods to create mimics stabling the prefusion trimer that presents critical NAb epitopes with appropriate fidelity [29]. Multi-mAb therapy is a promising approach to treating HIV infection and may be a candidate for SARS-CoV-2 [28]. It would allow for the bypassing of viral escape strategies and restoration of an adapted endogenous immune response. These antibodies inhibit the more effective, broadly cross-neutralizing antibodies (bnAbs), preventing the immune reaction from progressing.

### 2.4. Designing Future Vaccines with Glycosylation in Mind

The most utilized and internationally accepted SARS-CoV-2 vaccines are nucleic acid vaccines such as the Pfizer-BioNTech and the Moderna Spikevax. For RNA-based mRNA vaccines, the mRNA is transcribed in vitro from DNA before the RNA is encapsulated in the nanoparticles prior to vaccine administration, so no DNA is present, and no transcription is needed in the nucleus [30,31,32,33].

While these vaccines are some of the most efficacious available, there is potential for host glycosylation to negatively impact their success. Previous studies have found that host protein glycosylation impairs immune responses against bacteria even after using nucleic acid vaccines [34]. Similar findings are seen in viruses where glycosylation changes the structure and efficacy of the mRNA-derived viral antigens through post-translational modifications [30]. One proposed theory for this phenomenon is that glycans shield the protein surface so that the protein folding deviates from its native form [30] (Figure 1B). The receptor binding domain (RBD) of the SARS-CoV-2 S protein, which is responsible for recognizing and binding to the ACE2 receptor, is found in a closed or inaccessible receptor conformation, preventing it from being recognized by immune cells [35]. However, the receptor binding motif (RBM), the most crucial component for attachment to the ACE2 receptor, remains untouched and stable [35]. Glycan masking is also implicated in this immune shifting, or a shift in the immune response away from specific epitopes while focusing on other epitopes to decrease the extent of off-target antibody production. This also has been linked with reduced ADE activity and increased B cell regulation and testing [4]. However, there are some downsides to glycan masking. The process is complicated and requires extensive rounds of glycosylation in the correct sites, or it may lead to misfolding and off-target downstream effects [4].

A new approach introduces specific glycosylation sites onto the RBD to counteract this issue and thus elicit more robust immune responses to both previous and mutated variants. A novel study by Carnell et al. proposed a modified RBD spike subunit that triggers more broad neutralizing immune responses as a potentially improved booster vaccine [35]. Their methods were based on studies conducted with MERS and influenza viruses and involved removing and adding new glycan sites. In preliminary results, they demonstrated that this method could produce more neutralizing antibodies when compared to wild-type SARS-CoV-2 RBD.

Moreover, data suggests that the N-glycosylation and O-glycosylation of proteins can occur [30]. This glycosylation has been shown to impact NAb epitopes by creating steric hindrance that prevents proper antibody responses and has also been implicated in the recruitment of T cells [30]. Targeting these N-glycosylation sites may prove to be beneficial in creating more efficient vaccines that can withstand several mutations; however, in practice, it is more complex. There are 22 potential N-glycoSites on the S protein of SARS-CoV-2, which, as Huang et al. highlighted, make the process of analysis and vaccine design more difficult [36]. The role of N-glycan epitopes extends beyond the recruitment of immune cells; they are crucial for viral entry by mediating membrane fusion. This is seen in the HIV-1 virus, where the densely packed N-glycans on the viral envelope are responsible for pathogenesis [37]. This concept has also been studied in relation to other pathogens, such as influenza and Ebola [30].

From the SARS-CoV-2 pandemic, we can learn a lot about vaccine design and efficacy. It is evident that glycosylation plays an essential and contrasting role in immunogenicity; therefore, it has been suggested that removing the glycosylation sites around vulnerable NAb epitopes may be an effective strategy. The ACE2 receptor binding domain of the SARS-CoV-2 spike protein, which is required for viral anchoring to enable cell entry, is not glycosylated to increase the fitness of the virus [38]. However, the other epitopes in the S protein are glycosylated, which has been suggested to divert the vaccine-induced immune response.

Another method that may improve the efficacy of these nucleic acid vaccines is to create proper glycosylation matches. This can be accomplished by ensuring that the same cell types infected during viral infection express the nucleic acid vaccines. The rationale for this approach is based on the fact that the surface glycoproteins of enveloped viruses use the host glycosylation machinery, and the glycosylation of the viral proteins must be highly similar to the glycosylation of the mRNA vaccines [38]. Adopting the S1 antigen instead of the whole S protein is another potential suggestion for improving these nucleic acid vaccines. In this way, the immune response to these vaccines is targeted not only in the virion phase but also in the activation phase. This activation phase frees the ACE2-anchoring S1 subunit, providing more opportunities for the vaccine to expose the NAb epitopes to the immune system and prevent immune evasion [38]. While these suggestions may improve the immune response to the SARS-CoV-2 virus, other factors should be considered. Namely, host inflammation has been shown to alter the glycosylation process and must be considered when designing new vaccines [39]. This can be a detrimental factor in vaccine design, as a key feature of COVID-19 is a cytokine release syndrome (CRS) that leads to a prolonged inflammatory phase. Some of the pro-inflammatory cytokines upregulated in COVID-19 patients include TNF-α, IL-6, IL-8, IL-17, and IL-1β, which are responsible for the increased ability of cells to respond to the infection by increasing the recruitment of neutrophils, macrophages, and T lymphocytes and increasing vascular permeability [40]. The downside of CRS is acute injury to the lung, microbiota alteration, and other damage to cellular and organ functions seen in many patients who have previously experienced severe COVID-19 symptoms [40]. As this is a concern for most patients, identifying glycosylation sites for vaccine design may prove even more difficult than anticipated. In other non-SARS-CoV-2 diseases, the induction of pro-inflammatory cytokines leads to conformational changes in the N-glycosylation of endothelial cells [39]. In reverse, glycosylation can also impact inflammatory responses by regulating the recruitment of leukocytes [39], demonstrating that the relationship between glycosylation and inflammation is a directly correlation. There is a gap in the literature discussing this relationship in terms of SARS-CoV-2 infection; however, our knowledge of the exact impact of inflammation and glycosylation on each other will be useful when designing vaccines for future COVID-19 variants and may even be potentially translatable to future pandemics.

## 3. Conclusions

While the COVID-19 pandemic may seem to be a part of history, it presents a great learning opportunity for preparing for a potential future pandemic of the exact same nature. If vaccines are produced and used for the new Omicron variants, how will we know that future mutated variants will not cause the same problem again? Since the probability of such a problem occurring is very low, evidence based on the existing data indicates that the cross-reactivity of antibodies is a phenomenal concept that can either be advantageous or detrimental to the host [41]. For example, it is possible that the antibodies produced against the recent subvariants of Omicron may, for the future subvariant, have a helpful cross-reactive reaction on the N-terminal domain epitopes of the S trimer, which constitute the two primary neutralizing targets for the neutralizing antibody. This reaction would decrease the RBD binding affinity of SARS-CoV-2 to ACE2. So, it is logical to continue universal vaccination in order to prevent the cycle of multiplication and further mutation of the SARS-CoV-2 virus on the population level. At the same time, efforts should be made to develop new vaccines that target novel epitopes absent from the primary variant or to create new recombinant drugs that do not lose their effectiveness when the virus changes [42,43]. This knowledge can help to ensure that we are more prepared for the next pandemic with efficacious vaccines that may be developed quickly.

## 4. Patents

M.R.S. reports patents 55983477-9CN and CAN-DMS-150056368.1 on the compositions and methods for the treatment of coronavirus infection and respiratory compromise. M.R.S. reports having sought approval from Health Canada to test the therapeutic treatment reported in the patents in a human clinical trial.

## Figures and Tables

**Figure 1 viruses-15-02079-f001:**
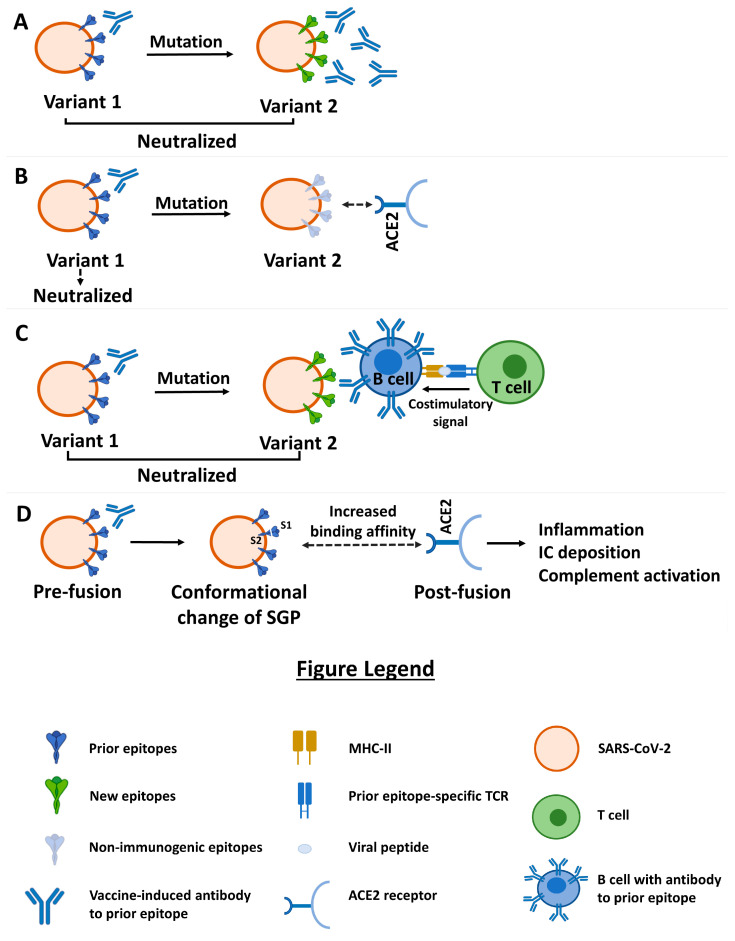
The various ways in which the immune system can still mount a response against mutated variants. (**A**) Despite the single mutation in the epitope of the second variant, the epitope-specific antibodies from the first variant can still bind and neutralize the new variant if the amount of antibodies is high enough from a booster injection. (**B**) The SARS-CoV-2 spike epitope may be converted into a non-immunogenic form after mutation, allowing the binding of the S protein to the ACE2 receptor. (**C**) The new epitope-specific B cell presents the prior epitope to the helper T cell with the aid of a co-stimulatory signal from the prior epitope-specific memory helper T cell. (**D**) Mutated variants may have increased affinity to bind to ACE2 due to conformational changes in the SPG through FcR-independent antibody-dependent enhancement (Fc-independent ADE).

## Data Availability

Data are contained within the article.

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
