# Peer review of "Contemporaneous SARS-CoV-2-Neutralizing Antibodies Mediated by N-glycan Shields"

_viruses, 2023, doi:10.3390/v15102079_

Round 1
Reviewer 1 Report
This is a perspective manuscript to describe contemporaneous SARS-C0V-2 neutralizing antibodies mediated by decotope N-glycan shields. Different vaccine platforms for SARS-CoV-2 against the origin and variant virus, and natural infections provide a better understanding on immune response by B and T cells interaction. However, this perspective is more as a review article. For example, the authors suggested designing vaccines with glycosylation to broadly enhance immune response in future but discussed only based on several publications. This is lack of own opinion and own design.
Author Response
Reviewer # 1:
Comments and Suggestions for Authors
This is a perspective manuscript to describe contemporaneous SARS-C0V-2 neutralizing antibodies mediated by decotope N-glycan shields. Different vaccine platforms for SARS-CoV-2 against the origin and variant virus, and natural infections provide a better understanding on immune response by B and T cells interaction. However, this perspective is more as a review article. For example, the authors suggested designing vaccines with glycosylation to broadly enhance immune response in future but discussed only based on several publications. This is lack of own opinion and own design.
Submission Date19 August 2023 Date of this review04 Sep 2023 06:05:22
Author response:
Thank you for your comments. Upon further review as well as other suggestions by the other reviewers we have decided to change this from a perspective to a review article. While our initial goal was to share our opinion while also backing it up using literature, it seems the end result was more a review article, therefore we have changed it to be as such.
Reviewer 2 Report
The article is devoted to the important issue of immune evasion of new viral variants from natural infection- or vaccine-induced antibodies.
Of particular interest to the article is the consideration of Thomas Francis's theory, which received the figurative name Original Antigenic Sin. As stated in Abstract “This paper presents some convincing immunologic reasons that may challenge this theory” (OAS). Unfortunately, I did not find in the text such “convincing immunologic reasons”.
The review does not show the main idea that the authors want to express. The text contains a large number of semantic repetitions. Compare for example Lines 102-105 and 113-116. I believe the review needs a major revision.
There are also some specific comments.
1.Figure 1 shows the pathway for new viral variants to emerge. I understand subpanel b, in which variant 2 is formed from WT. Variant 2 carries a mutated Spike. However, in subpanels a, c and d, viral particles simultaneously carry both WT and mutated Spikes. On subpanel a, variant 3 carries three types of Spike at once. Theoretically, the existence of hybrid virus particles is possible, but in practice their proportion is extremely small.
2.The term Decotope is used in the title of the article, once in Significance, and once again in the Introduction, where it is defined. Further in the article, the term Decotope is not used anywhere else. How justified was it to put this term in the title of the article and give it a definition?
3.Lines 152-154: Other evidence indicates that individuals vaccinated with some mRNA-based Alpha or Delta variant vaccines or naturally infected have a relatively decreased response to Omicron variant-specific epitopes compared to the unvaccinated individuals.
Do unvaccinated individuals have any kind of SARS-CoV-2 immunity?
4.Line 199: “mRNA directly transcribed in the host nucleus”.
I invite the authors to check this phrase.
5.Lines 258-259: the glycosylation of the viral proteins must be highly similar to the glycosylation of the mRNA vaccines [32].
It is true that the viral Spike and mRNA vaccine Spike share the same primary sequence. However, the type of glycosylation is highly dependent on the host cells in which it occurs. Viral Spike and the mRNA vaccine Spike can be expressed in different types of cells with different types of glycosylation. In support of this, the authors themselves make the following observation: “Namely, host inflammation has been shown to alter the glycosylation process and must be considered when designing new vaccines [32].”
6.The referencing in some places of the text is not entirely accurate.
For example, in Lines (146-150) the authors write about “increases their (Spike) binding affinity to ACE2”. In the cited articles [21] and [22], I did not find this information. These articles provide data only on the affinity of the interaction of antibodies and the Spike.
In Lines (152-154) the authors write: “individuals vaccinated with some mRNA-based Alpha or Delta variant vaccines or naturally infected have a relatively decreased response to Omicron variant-specific epitopes compared to the unvaccinated individuals [23].” I did not find any mention of variant specific vaccines in publication [23].
Author Response
Reviewer # 2
Comments and Suggestions for Authors
The article is devoted to the important issue of immune evasion of new viral variants from natural infection- or vaccine-induced antibodies.
Of particular interest to the article is the consideration of Thomas Francis's theory, which received the figurative name Original Antigenic Sin. As stated in Abstract “This paper presents some convincing immunologic reasons that may challenge this theory” (OAS). Unfortunately, I did not find in the text such “convincing immunologic reasons”.
The review does not show the main idea that the authors want to express. The text contains a large number of semantic repetitions. Compare for example Lines 102-105 and 113-116. I believe the review needs a major revision.
Author response:
- Thank you for the point. We have reviewed the lines highlighted and agree that there is some repetition. We have corrected it by removing some parts and combining others to avoid redundancies.
There are also some specific comments.
1.Figure 1 shows the pathway for new viral variants to emerge. I understand subpanel b, in which variant 2 is formed from WT. Variant 2 carries a mutated Spike. However, in subpanels a, c and d, viral particles simultaneously carry both WT and mutated Spikes. On subpanel a, variant 3 carries three types of Spike at once. Theoretically, the existence of hybrid virus particles is possible, but in practice their proportion is extremely small.
Author response:
- We acknowledge that the likelihood is low and will highlight this in text.
2.The term Decotope is used in the title of the article, once in Significance, and once again in the Introduction, where it is defined. Further in the article, the term Decotope is not used anywhere else. How justified was it to put this term in the title of the article and give it a definition?
Author response:
- The term decotope is appropriate in this paper, so we will be keeping the term and definition in the introduction, however, will remove it from the title.
3.Lines 152-154: Other evidence indicates that individuals vaccinated with some mRNA-based Alpha or Delta variant vaccines or naturally infected have a relatively decreased response to Omicron variant-specific epitopes compared to the unvaccinated individuals.
Do unvaccinated individuals have any kind of SARS-CoV-2 immunity?
Author response:
- Unvaccinated individuals have been found to have immunity against SARS-CoV-2 only if they had a prior infection. Unvaccinated individuals who were naturally infected had an 85% lower risk of getting COVID-19 than those who were not naturally infected. This was seen in both mild and severe disease cases.
Citation: Ridgway, J. P., Tideman, S., Wright, B., & Robicsek, A. (2022). Rates of COVID-19 among unvaccinated adults with prior COVID-19. JAMA Network Open, 5(4), e227650-e227650.
4.Line 199: “mRNA directly transcribed in the host nucleus”.
I invite the authors to check this phrase.
Author response:
- The source that is cited for that phrase states as follows: “mRNA is the intermediate genetic material that is produced (transcribed) from protein-coding DNA in the nucleus.”
Citation: Ozdilek, A., & Avci, F. Y. (2022). Glycosylation as a key parameter in the design of nucleic acid vaccines. Current Opinion in Structural Biology, 73, 102348.
5.Lines 258-259: the glycosylation of the viral proteins must be highly similar to the glycosylation of the mRNA vaccines [32].
It is true that the viral Spike and mRNA vaccine Spike share the same primary sequence. However, the type of glycosylation is highly dependent on the host cells in which it occurs. Viral Spike and the mRNA vaccine Spike can be expressed in different types of cells with different types of glycosylation. In support of this, the authors themselves make the following observation: “Namely, host inflammation has been shown to alter the glycosylation process and must be considered when designing new vaccines [32].”
Author response:
- Thank you for this comment. We agree with this comment.
6.The referencing in some places of the text is not entirely accurate.
For example, in Lines (146-150) the authors write about “increases their (Spike) binding affinity to ACE2”. In the cited articles [21] and [22], I did not find this information. These articles provide data only on the affinity of the interaction of antibodies and Spike.
In Lines (152-154) the authors write: “individuals vaccinated with some mRNA-based Alpha or Delta variant vaccines or naturally infected have a relatively decreased response to Omicron variant-specific epitopes compared to the unvaccinated individuals [23].” I did not find any mention of variant specific vaccines in publication [23].
Author response:
- The text has been edited to include the updated information. Thank you for pointing this out.
Reviewer 3 Report
The manuscript by Leili Baghaie et al., is a perspective view that may challenge the Original Antigenic Sin (OAS) theory, described in 1960 by Thomas Francis, and it suggests to continue universal vaccination to stop further mutation of the SARS-CoV-2 virus. At the same time the information presented in this perspective are useful in developing vaccines that target novel epitopes. These are interesting points that should be kept in mind to understand the immune responses to pathogens and vaccines. There is a point that should be addressed, on lines 119-123 the Authors state: “Infection with a highly glycosylated variant induces a deficient adaptive response with a lack of neutralizing antibodies and a robust T response. In the event of reinfection with a low-glycosylated variant, antibody-mediated neutralization is significantly reduced, and the disease is more severe”. This statement implies that although a robust T response is induced the disease is more severe. The Authors should better explain this point taking into account that a robust T cell response can mediate a better protection and leads to a less severe disease.
Author Response
Reviewer # 3:
Comments and Suggestions for Authors
The manuscript by Leili Baghaie et al., is a perspective view that may challenge the Original Antigenic Sin (OAS) theory, described in 1960 by Thomas Francis, and it suggests continuing universal vaccination to stop further mutation of the SARS-CoV-2 virus. At the same time the information presented in this perspective are useful in developing vaccines that target novel epitopes. These are interesting points that should be kept in mind to understand the immune responses to pathogens and vaccines. There is a point that should be addressed, on lines 119-123 the Authors state: “Infection with a highly glycosylated variant induces a deficient adaptive response with a lack of neutralizing antibodies and a robust T response. In the event of reinfection with a low-glycosylated variant, antibody-mediated neutralization is significantly reduced, and the disease is more severe”. This statement implies that although a robust T response is induced the disease is more severe. The Authors should better explain this point taking into account that a robust T cell response can mediate a better protection and leads to a less severe disease.
Author response:
Thank you for your comments. While CD4 T cells are vital in the promotion of antibody responses and induction of inflammation that normally reduces disease severity, in COVID-19 and other viral diseases that doesn’t seem like the case:
- “It therefore remains to be determined whether robust memory T cell responses in the absence of detectable circulating antibodies can protect against severe forms of COVID-19. This scenario has nonetheless been inferred from previous studies of MERS and SARS-CoV-1 (Channappanavar et al., 2014; Li et al., 2008; Zhao et al., 2016, 2017), both of which have been shown to induce potent memory T cell responses that persist for years, in contrast to the corresponding antibody responses (Alshukairi et al., 2016; Shin et al., 2019; Tang et al., 2011).”
Citation: Sekine, T., Perez-Potti, A., Rivera-Ballesteros, O., Strålin, K., Gorin, J. B., Olsson, A., ... & Buggert, M. (2020). Robust T cell immunity in convalescent individuals with asymptomatic or mild COVID-19. Cell, 183(1), 158-168.
“Despite the reported T cell lymphopenia in severe disease, SARS-CoV-2-specific CD4+ and CD8+ T cell frequencies increased with disease severity (Anft et al., 2020; Peng et al., 2020).”
Citation: Bacher, P., Rosati, E., Esser, D., Martini, G. R., Saggau, C., Schiminsky, E., ... & Scheffold, A. (2020). Low-avidity CD4+ T cell responses to SARS-CoV-2 in unexposed individuals and humans with severe COVID-19. Immunity, 53(6), 1258-1271.
Round 2
Reviewer 1 Report
The manuscript was improved in the revised version. Suggest to include the immune response and preliminary results of the 2nd bivalent COVID-19 vaccine booster in the text.
Author Response
Reviewer #1: Comments and Suggestions for Authors
The manuscript was improved in the revised version. Suggest to include the immune response and preliminary results of the 2nd bivalent COVID-19 vaccine booster in the text.
Author response: Thank you for the suggestion. We have added the following paragraph in the text, Lines 153-162:
Mattiuzzi and Lippi [25] provided a review on the analyses of COVID-19 vaccine efficacy in older persons who received the second booster compared to unvaccinated people and those receiving only a single COVID-19 vaccine booster. The second vaccine booster maintained high effectiveness against adverse COVID-19 outcomes such as hospitalization, intensive care unit admission and death (i.e., between 77 and 86%), and also showed around 10% higher efficacy than the single booster. The efficacy of the second vaccine booster declined over time, decreasing by 33–46% when assessed at >120 days from administration. The results of these ad interim analyses of the ongoing Italian nationwide COVID-19 vaccination campaign suggested that regular boosting with COVID-19 vaccines may be advisable in older persons [25].
25. Mattiuzzi, C.; Lippi, G. Efficacy of the Second COVID-19 Vaccine Booster Dose in the Elderly. Vaccines (Basel) 2023, 11, doi:10.3390/vaccines11020213.
Reviewer 2 Report
The authors presented a revised version of the manuscript, in which they took into account some of the reviewers’ comments. The changes made can be described as minor, but not major as recommended.
I do not think that that manuscript has been sufficiently improved to warrant publication in Viruses.
Some comments I made in the original review were not addressed in the revised version.
In particular
1.Figure 1 shows the pathway for new viral variants to emerge. I understand subpanel b, in which variant 2 is formed from WT. Variant 2 carries a mutated Spike. However, in subpanels a, c and d, viral particles simultaneously carry both WT and mutated Spikes. On subpanel a, variant 3 carries three types of Spike at once. Theoretically, the existence of hybrid virus particles is possible, but in practice their proportion is extremely small.
Author response:
We acknowledge that the likelihood is low and will highlight this in text.
I still disagree with Figure 1. The authors write that “The likelihood of having a hybrid (both WT and mutated) epitope is low but possible, therefore this is a theoretical diagram” (Lines 218-219). I don’t understand why it is necessary to present diagrams that have no relation to reality. Reviews are meant to clarify issues, not to confuse the reader.
3.Lines 152-154: Other evidence indicates that individuals vaccinated with some mRNA-based Alpha or Delta variant vaccines or naturally infected have a relatively decreased response to Omicron variant-specific epitopes compared to the unvaccinated individuals.
Do unvaccinated individuals have any kind of SARS-CoV-2 immunity?
Author response:
Unvaccinated individuals have been found to have immunity against SARS-CoV-2 only if they had a prior infection. Unvaccinated individuals who were naturally infected had an 85% lower risk of getting COVID-19 than those who were not naturally infected. This was seen in both mild and severe disease cases.
Citation: Ridgway, J. P., Tideman, S., Wright, B., & Robicsek, A. (2022). Rates of COVID-19 among unvaccinated adults with prior COVID-19. JAMA Network Open, 5(4), e227650-e227650.
To avoid any ambiguity, it is necessary to clarify that unvaccinated individuals were naturally infected.
4.Line 199: “mRNA directly transcribed in the host nucleus”.
I invite the authors to check this phrase.
Author response:
The source that is cited for that phrase states as follows: “mRNA is the intermediate genetic material that is produced (transcribed) from protein-coding DNA in the nucleus.”
Citation: Ozdilek, A., & Avci, F. Y. (2022). Glycosylation as a key parameter in the design of nucleic acid vaccines. Current Opinion in Structural Biology, 73, 102348.
Indeed, the paper Ozdilek, A., & Avci, F. Y. (2022) contains the phrase “mRNA is the intermediate genetic material that is produced (transcribed) from protein-coding DNA in the nucleus.” However, this phrase does not apply to mRNA vaccines. Further in the article Ozdilek, A., & Avci, F. Y. (2022) follows the phrase “mRNAs introduced in mRNA vaccines can be directly used by ribosomes for protein production.”
The authors write: “These vaccines (Pfizer-BioNTech and the Moderna Spikevax) comprise mRNA directly transcribed in the host nucleus” (Lines 184-185). This phrase is absolutely wrong. I cannot recommend a review containing such errors for publication in Viruses.
Author Response
- Figure 1 shows the pathway for new viral variants to emerge. I understand subpanel b, in which variant 2 is formed from WT. Variant 2 carries a mutated Spike. However, in subpanels a, c and d, viral particles simultaneously carry both WT and mutated Spikes. On subpanel a, variant 3 carries three types of Spike at once. Theoretically, the existence of hybrid virus particles is possible, but in practice their proportion is extremely small.
Author response: Thank you for the suggestion. We have added text in the figure legend to clarify this issue:
Figure 1. The various ways the immune system can still mount a response against mutated variants. a) Despite the single mutation in the epitope of the second variant, the epitope-specific antibodies from the first variant can still bind and neutralize variant-2 using the prior epitopes. However, further mutations are unable to bind prior epitope-specific antibodies and will have access to ACE2 and the new epitopes specific naïve B cells. On subpanel a, variant 3 carries three types of Spikes at once. Theoretically, the existence of hybrid virus particles is possible, but in practice their proportion is extremely small. The likelihood of having a hybrid (both WT and mutated) epitope is low but possible. b) The SARS-CoV-2 spike epitope may convert into a non-immunogenic form after mutation; c) The new epitope-specific B cell presents the prior epitope to the helper T cell with the aid of a co-stimulatory signal from the prior epitope-specific memory helper T cell; d) Mutated variants may have increased affinity to bind to ACE2 using antibodies specific to prior (pre-mutated) epitopes through FcR-independent antibody-dependent enhancement (Fc-independent ADE).
2. Lines 152-154: Other evidence indicates that individuals vaccinated with some mRNA-based Alpha or Delta variant vaccines or naturally infected have a relatively decreased response to Omicron variant-specific epitopes compared to the unvaccinated individuals. Do unvaccinated individuals have any kind of SARS-CoV-2 immunity?
Author response: Thank you for the suggestion. We have added the following to the text to clarify this issue.
Other evidence indicates that individuals vaccinated with antigens against Wuhan-Hu-1 and then infected with Alpha or Delta variants have a relatively decreased response to variant-specific epitopes compared to the unvaccinated individuals which were naturally infected [22]. Unvaccinated individuals have been found to have immunity against SARS-CoV-2 only if they had a prior infection. Unvaccinated individuals who were naturally infected had an 85% lower risk of getting COVID-19 than those who were not naturally infected [23]. This was seen in both mild and severe disease cases.
23. Ridgway, J.P.; Tideman, S.; Wright, B.; Robicsek, A. Rates of COVID-19 Among Unvaccinated Adults With Prior COVID-19. JAMA Netw Open 2022, 5, e227650, doi:10.1001/jamanetworkopen.2022.7650.
3. Indeed, the paper Ozdilek, A., & Avci, F. Y. (2022) contains the phrase “mRNA is the intermediate genetic material that is produced (transcribed) from protein-coding DNA in the nucleus.” However, this phrase does not apply to mRNA vaccines. Further in the article Ozdilek, A., & Avci, F. Y. (2022) follows the phrase “mRNAs introduced in mRNA vaccines can be directly used by ribosomes for protein production.”
The authors write: “These vaccines (Pfizer-BioNTech and the Moderna Spikevax) comprise mRNA directly transcribed in the host nucleus” (Lines 184-185). This phrase is absolutely wrong. I cannot recommend a review containing such errors for publication in Viruses.
Author response: Thank you for pointing this misprint out for us. We have added the following to the text to correct this issue:
The most utilized and internationally accepted SARS-CoV-2 vaccines are nucleic acid vaccines such as the Pfizer-BioNTech and the Moderna Spikevax. These vaccines comprise mRNA directly transcribed from protein-coding DNA in the nucleus to encode the viral antigenic protein [29]. mRNAs introduced in mRNA vaccines can be directly used by ribosomes for protein production. Compared to traditional protein vaccines that mainly activate antibody production, the mRNA vaccines stimulate both humoral and cellular immune responses [30]. There are no other risks for mRNA vaccines applications compared to recombinant techniques [31]. In comparison with DNA vaccines, mRNA vaccines are just delivered to cytoplasmic region of the cell, eliminating the risk of genomic integration [31,32]. The mRNA vaccines can directly lead to high velocity immunogen production without necessity of crossing the nuclear membrane barrier, so, their expression is unrestricted in the packing step [32].
- Ozdilek, A.; Avci, F.Y. Glycosylation as a key parameter in the design of nucleic acid vaccines. Curr Opin Struct Biol 2022, 73, 102348, doi:10.1016/j.sbi.2022.102348.
- Schlake, T.; Thess, A.; Fotin-Mleczek, M.; Kallen, K.J. Developing mRNA-vaccine technologies. RNA Biol 2012, 9, 1319-1330, doi:10.4161/rna.22269.
- Wadhwa, A.; Aljabbari, A.; Lokras, A.; Foged, C.; Thakur, A. Opportunities and Challenges in the Delivery of mRNA-based Vaccines. Pharmaceutics 2020, 12, doi:10.3390/pharmaceutics12020102.
- Yamamoto, A.; Kormann, M.; Rosenecker, J.; Rudolph, C. Current prospects for mRNA gene delivery. Eur J Pharm Biopharm 2009, 71, 484-489, doi:10.1016/j.ejpb.2008.09.016.
Round 3
Reviewer 2 Report
I still think, that Figure 1 is wrong. Changing the legend does not correct this Figure.
I still cannot agree with the authors' statement that “These vaccines (Pfizer-BioNTech and the Moderna Spikevax) comprise mRNA directly transcribed from protein-coding DNA in the nucleus to encode the viral antigenic protein [29].” (Lines 196-198). The reference to the paper [29] is inappropriate. The paper [29] makes no such claim.
Thus, compared to revised version #2, the authors did not make significant changes, so my opinion remains the same. I believe that the manuscript as presented should not be accepted.
Author Response
I still think, that Figure 1 is wrong. Changing the legend does not correct this Figure.
Author response: We have revised the Figure 1 according to the reviewer's comment. Thank you for highlighting this issue.
I still cannot agree with the authors' statement that “These vaccines (Pfizer-BioNTech and the Moderna Spikevax) comprise mRNA directly transcribed from protein-coding DNA in the nucleus to encode the viral antigenic protein [29].” (Lines 196-198). The reference to the paper [29] is inappropriate. The paper [29] makes no such claim.
Author response: We have removed the sentence and cited reference from the revised manuscript: "I still cannot agree with the authors' statement that “These vaccines (Pfizer-BioNTech and the Moderna Spikevax) comprise mRNA directly transcribed from protein-coding DNA in the nucleus to encode the viral antigenic protein [29].” (Lines 196-198). The reference to the paper [29] is inappropriate. The paper [29] makes no such claim.
Thus, compared to revised version #2, the authors did not make significant changes, so my opinion remains the same. I believe that the manuscript as presented should not be accepted.
Author response: We made made the corrections as suggested by the reviewer #2. The other two peer-reviewers recommended publications.
Thus, compared to revised version #2, the authors did not make significant changes, so my opinion remains the same. I believe that the manuscript as presented should not be accepted.
Author response: We hope that you will kindly accepted our corrections and revisions.